# Clean H_2_ Production by Lignin-Assisted Electrolysis in a Polymer Electrolyte Membrane Flow Reactor

**DOI:** 10.3390/ma16093525

**Published:** 2023-05-04

**Authors:** José-Enrique Rodríguez-Fernández, María Rojo, Juan Ramón Avilés-Moreno, Pilar Ocón

**Affiliations:** Departamento de Química Física Aplicada, Universidad Autónoma de Madrid (UAM), C/Francisco Tomás y Valiente 7, 28049 Madrid, Spain; joseenrique.rodriguez@estudiante.uam.es (J.-E.R.-F.); maria.rojoc@estudiante.uam.es (M.R.); pilar.ocon@uam.es (P.O.)

**Keywords:** green H_2_, lignin, electrochemistry, additives, assisted-electrolysis, physicochemical characterisation, flow cell, AEM electrolyser

## Abstract

Biomass-derived products, such as lignin, are interesting resources for energetic purposes. Lignin is a natural polymer that, when added to the anode of an alkaline exchange membrane water electrolyser, enhances H_2_ production rates and efficiencies due to the substitution of the oxygen evolution reaction. Higher efficiencies are reported when different catalytic materials are employed for constructing the lignin anolyte, demonstrating that lower catalytic loadings for the anode improves the H_2_ production when compared to higher loadings. Furthermore, when a potential of −1.8 V is applied, higher gains are obtained than when −2.3 V is applied. An increase of 200% of H_2_ flow rates with respect to water electrolysis is reported when commercial lignin is used coupled with Pt-Ru at 0.09 mg cm^−2^ and E = −1.8 V is applied at the cathode. This article provides deep information about the oxidation process, as well as an optimisation of the method of the lignin electro-oxidation in a flow-reactor as a pre-step for an industrial implementation.

## 1. Introduction

Fossil fuels have been the cornerstone for society development and economic growth. In exchange, the combustion of these energy sources has generated polluting products and lead to an increase in the global CO_2_ concentration. Therefore, the development of new and renewable energy sources is key for the maturity of a sustainable society [1]. In this context, H_2_ is an essential raw material for many industries, such as steel, biofuels, and ammonia industries [2,3,4]. It is a possible fuel solution for decarbonisation and a form of energy storage resource. Moreover, its energy density is 2.75 times higher than classic hydrocarbon combustibles [5,6]. There are different forms for the production of H_2_. The most traditional ways are the coal gasification, the natural gas reforming, and the partial oxidation of hydrocarbons. Although these production methods have been optimised over the years, the most ideal route is the electrolysis of water using a renewable energy source, with the main intention of reducing CO_2_ emissions [7]. Furthermore, there is a need to invest in the development of water electrolysis with alkaline exchange membrane (AEM). Despite the low currents required for alkaline electrolysis, the productivities are comparably better due to the cheaper catalysts used in this technology, in comparison with the ones used in proton exchange membrane (PEM) water electrolysis [8].

In recent years, several advances have been made in AEM electrolysis. Recently, new catalysts [9,10,11], such as stainless steel and Ni-Fe alloys, are being studied for this purpose. The development of new membranes [12,13] and different techno-economic studies [14,15] generally show a clear desire of the scientific community to advance in the AEM process for the production of H_2_.

The electrochemical production of H_2_ by water splitting involves the fulfilment of two reactions. The first is the hydrogen evolution reaction (HER), which occurs relatively fast and without any remarkable kinetic impediment. The second is the oxygen evolution reaction (OER), which many authors describe [16,17,18,19,20] as a reaction with sluggish kinetics that becomes the rate-determining process limiting the H_2_ production. Over the years, there have been many research projects on catalysts for OER, such as Ru, Ir, and Pt nanoparticles or Fe-based catalysts, for the enhancement of this reaction, and thus the increase in its kinetic rates [16,17,18,19,20]. The use of organic molecules (alcohols, biomass, or black-liquor products) added to the anode reduces the overpotential of the process and increases its efficiency. The comparatively lower anodic overpotential, and therefore higher kinetic rates presented by these materials, justifies the studies carried out for this purpose [6,21,22,23,24,25,26,27,28,29,30,31,32,33].

Biomass materials are the keystone for a more sustainable development, in which the revaluation of waste by-products contributes to the growth of energy storage technologies. These materials present the ability to reduce the overpotential for H_2_ production. Cellulose, hemicelluloses, and lignin are the main components of the plant cell wall, linked together by a highly ordered matrix called lignocellulose. In fact, lignins have been studied for the fulfilment of this goal [34,35,36,37,38,39,40,41,42,43,44].

Lignin is a natural polymeric bioproduct of the paper industry obtained after the treatment of black liquor. It is reported that the annual production of lignin is about 50–70 million tons, in which only around 1–2% of this quantity is leveraged for producing value-added products [45]. Figure 1 shows the three principal phenylpropanoids that constitute the lignin polymer. Its structure is irregular and without a defined and ordered form. The conjugation of these three monomers depends on the synthetic route that is undergone by the plant to reach the final material, in which the “Shikimate pathway” is involved in the polymer formation. The percentage of each monomer varies with the lignin source, and with the extraction plant [46,47,48]. Moreover, the way lignin is extracted from the black liquor will modify the final structure of the polymer obtained [49,50,51,52].

Many authors have proved the use of lignin as an additive in alkaline processes for the anode of an electrolyser. In fact, Caravaca et al. [37] have used a flow cell with Pt/C and Pt-Ru/C as catalysts in the cathode and anode, respectively. It is reported by the authors that H_2_ can be achieved at the cost of a high reduction in cell overpotential, in which E = 0.45 V or higher is applied. Moreover, lignin electrolysis is favoured in comparison with conventional electrolysis. Lalvani and Rajagopal [35] reported a linear relation between the lignin concentration and the current response when a Pt mesh is used as an electrode. Thereafter, both authors demonstrated that the use of this additive (i) leads to a more efficient H_2_ production, and (ii) that benzaldehydes are obtained as major products [39]. Other works have described the use of Ni as a catalyst for lignin electro-oxidation. For instance, Khalid et al. [40] have used Ni foam to perform a study on the effect of the flow rate on the current response in comparison with water electrolysis. Beliaeva et al. [36] employed Ni nanoparticles to explore the oxidation in an AEM electrolyser with lignin material and a model molecule (2-phenoxyethanol).

Innovative achievements are presented in this work, in which a thorough gas analysis of the gas products after lignin oxidation is performed. Furthermore, a new asymmetric set-up for catalysts is presented: Ni foam is used for the cathodic reaction, whereas Pt/C and Pt-Ru/C nanoparticles are the catalysts tested for the anodic counterpart. A complete analysis is carried out using two different loadings for each catalyst in lignin electro-oxidation. These loadings are lower in comparison with those used by Caravaca et al. [37]. These authors only used Pt-Ru for lignin electrolysis at a loading of 2 mg cm^−2^, whereas in this work, 0.09 and 0.89 mg cm^−2^ loadings are used for the Pt-Ru catalyst showing remarkable results for very low loadings in the anode. This original implementation combines the reported benefits of Ni over the HER [53], with the tests carried out in this work at different loadings for Pt/C and Pt-Ru/C over lignin electrolysis. In addition, fixed current tests have not been reported yet in bibliography for this type of setup and process to our knowledge. A flow reactor is used due to the willingness of having a better industrial approach to this process, and to demonstrate the applicability of this method to higher scales. Finally, electrolyte analysis is exhibited as an important cause of corrosion problems in these kinds of processes.

## 2. Materials and Methods

### 2.1. Materials

#### 2.1.1. Catalysts

Two different catalysts have been used in this project. Platinum nanoparticles (40%wt of Pt from Alfa Aesar, Haverhill (MA), USA Alfa Aesar) and multicomponent nanoparticles of Pt-Ru (30%wt Pt: 15%wt Ru from Alfa Aesar). The catalytic inks are prepared by adding 1 mg of the Pt or Pt-Ru nanoparticles and 20 μL of Nafion (solution at 10% *v*/*v*) to 1000 μL of a base solution (70–30 (*v*/*v*) isopropanol/water).

Three-electrode cell tests: This ink is loaded in a glassy carbon electrode (with a geometrical area of 0.07 cm^2^) by drop casting to a loading of 0.11 mg cm^−2^ of Pt and 0.12 mg cm^−2^ of Pt-Ru.

Flow cell tests: The ink composition is water-free to make the drying process easier and faster. The base solution is prepared with 6 mL of isopropanol and 100 µL of Nafion (solution at 10% *v*/*v*). Moreover, two different catalytic loadings, per catalyst, were tested in this new system. First, the catalyst loadings are 3.20 and 48.00 mg for Pt nanoparticles and 4.70 and 42.70 mg for Pt-Ru nanoparticles (including both elements). Thereafter, the ink is sprayed with a manual airbrush over a C cloth (with a geometrical area of 16 cm^2^) (purchased from Quintech^®^, Göppingen, Germany). Second, the catalyst loadings are 0.08 and 0.75 mg cm^−2^ for Pt and 0.09 and 0.89 mg cm^−2^ for Pt-Ru. Thereafter, the catalytic cloth is placed over a temperature-controlled heating plate at 60 °C for minutes to accelerate the drying process.

#### 2.1.2. Membrane

Fumasep FAA-3-50 membrane was used in the tests realised at the flow cell. The only step carried out was to immerse this membrane in NaOH 1 M solution for 24 h.

#### 2.1.3. Lignins

Two different lignin materials were used. One of them is commercial (from Sigma Aldrich^®^, Darmstadt, Germany), whereas the other was obtained by the ICP-CSIC Institute from biomass revalorisation processes, and hereinafter referred to as Pruning Biomass Lignin (Pruning B. Lignin) [54,55].

#### 2.1.4. Equipment

An Autolab^®^ PGSTAT302N potentiostat/galvanostat was used for the electrochemical measurements. An infrared spectrometer PerkinElmer^®^ SpectrumTwoTM (Waltham, MA, USA), coupled with an attenuated total reflection (ATR) accessory and UV-Vis spectra PerkinElmer^®^ Lambda 365 spectrophotometer were used for the characterisation of lignins. For the gas determination, a mass spectrometer (MS) Pfeiffer Vacuum Hi-Cube^®^ (Tecnovac, Alcobendas (Madrid), Spain) coupled with a gas chromatograph Varian 3900 were used, which had a Carboxen^®^-1006 PLOT GC column (purchased from Sigma Aldrich^®^). For the establishment of a regular temperature, a water bath and a temperature controller Eurotherm 2408 were used.

### 2.2. Methods

#### 2.2.1. Solubility Test

Lignin solubility was tested at different pH to analyse an optimum operational condition. In addition, this test was required due to the use of the non-commercial Pruning B. Lignin, which was a new material. Moreover, knowing its solubility characteristics was of interest. A concentration of 1 g L^−1^ was selected for this experiment. The alkaline pH selected were 8, 9, 10, 12, and 14. Furthermore, the feasibility of neutral pH and H_2_SO_4_ 10^−1^ M solubilisation were checked.

#### 2.2.2. Viscosity Measurements

Viscosity was measured with an Ostwald viscometer (COMECTA S.A.). Pure water, 1 M of NaOH solution, and 6 g L^−1^ of both lignin solutions in NaOH 1 M were measured.

#### 2.2.3. FT-IR with ATR Characterisation

Infrared spectra of the lignins were registered for solubilised and dry materials. The studied lignins were dissolved in NaOH 1 M to a final solution of 6 g L^−1^ of lignin. The experimental conditions were 16 scans and 2 cm^−1^ of resolution. The spectral range was selected between 4000 and 450 cm^−1^. The samples were analysed before and after oxidation (by performing the chronopotentiometry (CP) test at −100 mA cm^−2^ and 60 °C for 15 min. In these cases, the catalyst used was Pt at 0.08 mg cm^−2^).

#### 2.2.4. Mass Spectrometry Coupled with Gas Chromatography

A gas syringe was used for collecting a sample from the gas traps and introducing it into the chromatograph. Previously, a calibration curve was prepared with different gases; CO, CO_2_, and H_2_, using Ar as diluter gas. Interpolating the intensities registered by the mass spectrometer into the calibration curved determines the proportion of the different gases produced.

#### 2.2.5. NMR Characterisation

NMR spectra of lignin solutions before and after oxidation were obtained. The oxidised lignins were analysed after realising a CP test at −100 mA cm^−2^ and Pt at 0.08 mg cm^−2^ as catalyst. Water is the solvent used for the lignin solution, but the lignin solutions were diluted in D_2_O to improve the ^1^H-NMR signals. Taking advantage of the high concentration of lignin solutions, the dilution will permit the signal acquisition of the functional groups while keeping the water noise at minimum.

#### 2.2.6. UV-Vis Spectroscopy Characterisation

UV-Vis spectra of the NaOH anolyte were registered before and after three CP tests at −100 mA cm^−2^, 60 °C, 300 s, and Pt at 0.08 mg cm^−2^ as catalyst.

#### 2.2.7. Experimental Assembly

First, a three-electrode cell was used to check for the efficiency of the catalysts to oxidise the lignin materials. The reference electrode was an Ag/AgCl immersed in saturated KCl solution. The Pt wire was the counter electrode, and the working electrode was a glassy carbon loaded with the ink made of the nanoparticulated catalyst. Finally, these experiments were reproduced in a more industrial approach by performing them in a polymer electrolyte membrane reactor flow cell (Electrochem^®^, Bryan (TX), USA). This cell consists of two graphite plates, where the electrodes are placed and the cathode and anode are set up, separated by a polymeric membrane (dimensions of 4 × 4 cm^2^). The complete scheme of the cell is shown in Figure 2b.

The two catalysts were tested for the anodic reaction of the electrolyser. For the cathodic reaction, the hydrogen evolution reaction (HER), Ni foam was used in all tests. Cathode and anode were fed in a first instance with a solution of NaOH 1 M. Thereafter, to observe the effect produced by lignin on the electrolysis, the anode was fed with a lignin solution at 6 g L^−1^. The volume of the reservoirs was 100 mL and the solutions were operating in a recirculation mode.

The tubing system comprises the connection from the exits of the cell to two gas traps, one for the cathode and the other for the anode. The complete set up is shown in Figure 2a. The purpose is to accumulate the gas products in a time-controlled CP or chronoamperometry (CA). As a result, the volume of the generated gas can be measured and compared with the theoretical value obtained by the Faraday’s law. Moreover, the gases from the anode and cathode can be stored for analysis by mass spectrometry coupled with gas chromatography.

#### 2.2.8. Electrochemical Measurements

Electrochemical characterisation was performed by cyclic voltammetry (CV) in a three-electrode cell. Lignin solutions, at 6 g L^−1^ in NaOH 1 M, were compared to NaOH 1 M. The potential window selected was (−1 and 0.6 V vs. Ag/AgCl/KCl (sat)) for Pt and Pt-Ru catalysts. Chronoamperometries (CAs) at 0.2, 0.4, and 0.6 V were also performed.

Thereafter, a polymer electrolyte membrane flow reactor was used. In the new experimental assembly, only two electrodes were used in this cell. A Ni foam of 2 × 2 cm^2^ was used as the cathode, whereas the anode was comprised of a sprayed carbon cloth (2 × 2 cm^2^) with each catalyst (Pt and Pt-Ru) used at different catalytic loadings. CAs were performed at −1.8 and −2.3 V and CP at −100 mA cm^−2^. The gas obtained at the anode and cathode was trapped and measured. A mass spectrometric study was carried out only for the CP gas products with the willingness to know the gas composition.

## 3. Results and Discussion

### 3.1. Solubility Tests

Lignin samples were soluble in all alkaline pH solutions used. At neutral pH, solubilisation of the materials was also achieved, but this did not occur at acidic pH, in which both lignin samples were not soluble. As described in bibliography, kraft lignins are significantly more soluble in basic than in acid media [56,57]. It is important to note that the Pruning B. Lignin presents less capacity for solubilisation in comparison with the commercial one. This first material was needed to be ground with a mortar in order to avoid problems with the solubilisation process.

### 3.2. Viscosity Measurements

Viscosity is one of the parameters that has been demonstrated to affect the results obtained by the electrolysis in a flow cell [58]. Higher viscosities are undesired due to the higher energy required to impulse the flow of the electrolytes, and its impact on the costs increase due to lower pump efficiencies. Figure 3 shows the viscosities transformed into percentage in comparison with the water value for the lignin samples. Pruning B. Lignin was the most viscous solution (53 ± 8%), followed by the commercial lignin solution (34 ± 3%) and NaOH solution (24 ± 1%).

### 3.3. Electrochemical Characterisation

#### 3.3.1. Three-Electrode Cell

In Appendix A, it can be seen that the oxidation of lignin solutions is favoured compared to the evolution of oxygen in NaOH solutions. In addition, the reduction area of the Pt oxides decreases due to the effect of the lignin on the cathodic scan. Moreover, lignin diminishes the hydrogen adsorption area on the electrode surface (between −0.6 and −1 V). In fact, looking in detail at Appendix A, when Pruning B. Lignin is studied, it seems to be significantly more adsorbed on the Pt surface in comparison with the commercial lignin. Similarly, this effect can be appreciated in Pt-Ru (Appendix A) as a higher reduction in the hydrogen adsorption area.

Furthermore, Appendix A allows us to observe that the anodic branch of the lignin oxidation is more favourable than oxygen evolution. The lignin anodic sweep provides a higher current density response in comparison with the one obtained without lignin at the anode. Moreover, as it is known, biomass [6,26,29] and organic products present a lower overpotential for its oxidation in comparison with the OER.

Specifically, in regard to lignin oxidation and its anodic scan, it seems that for both organic materials and using Pt as catalyst, oxidation may present two peaks. The first is common for both lignins, and is located around −0.35 V. Commercial lignin presents another oxidation peak at around 0 V, whereas the Pruning B. Lignin presents the second oxidation peak at around 0.18 V. This effect is not seen at Pt-Ru as catalyst, where it is only appreciated at the higher current densities obtained for the lignin anodic scan, in comparison with the lower NaOH anodic scan current densities.

#### 3.3.2. Chronoamperometry

A brief analysis of the effect of lignin on the electrolysis was achieved by CA in a three-electrode cell (see Figure 4), where the enhancement of the process by the increase in the limit current density, in comparison with the one obtained by the NaOH solution, can be demonstrated. The advantage presented by the Pruning B. Lignin over the commercial material is shown for all the potentials studied (0.2, 0.4, and 0.6 V). Undoubtably, both lignins present a clear increase in their current density, denoting that the lignin oxidation reaction (LOR) occurs when lignin solutions are included for the electrolysis and partially inhibit the OER.

Apparently, Pruning B. Lignin shows higher current densities when Pt is used as catalyst (Figure 4a–c). In Appendix A, a connection with the results can be found, where at the CV, the Pruning B. Lignin presents significantly more current densities from 0 V to nearly 0.5 V, in comparison with the results obtained for the commercial lignin. Moreover, E = 0.2 V and E = 0.4 V match with the oxidation peaks observed for the Pruning B. Lignin at the CV (Appendix A). At 0.6 V, this high difference is not observed for the limiting current density obtained, compared to both lignins with the variations observed at the other conditions studied.

For Pt-Ru (Figure 4d–f), at 0.2 and 0.4 V, this distinction with the results obtained for both lignins is not observed. However, at 0.6 V, Pruning B. Lignin gives a higher current density in contrast to the one obtained for commercial lignin. Probably, this occurs through an activation of the oxidation process of the organic material by unknown processes not seen at the CV.

### 3.4. Polymer Electrolyte Membrane Reactor Flow cell

The prevalence of the LOR over the OER is due to the fact that it was demonstrated at the three-electrode cell. In the flow cell, CA and CP tests were carried out. Figure 5 shows the response in CP with both different catalysts and catalytic amounts. Lignin solutions present less absolute potential, denoting a more efficient process, around 100–200 mV lower when lignin anolyte is used.

However, hydrogen production does not entail a great difference between commercial and Pruning B. Lignin. This can be appreciated in Figure 6a, where the total hydrogen produced is shown (i.e., the total amount of the gas produced at the cathode and the amount of H_2_ which crosses the membrane to the anode due to crossover). A clear increase in H_2_ production, when lignin is used as anolyte, can be observed. Furthermore, it can be seen how the lignin solution inhibits the crossover phenomena, as determined by mass spectrometry. As hypothesis, the higher viscosity of lignin solutions might be the possible explanation for the decrement on the H_2_ crossover. Despite having a greater hydrogen production with lignin solutions, the volume crossing through the membrane is lower in comparison with NaOH without lignins. It can be hypothesised that a higher production at the cathode should imply a higher volume that could pass through the membrane. Moreover, it can be seen how Pruning B. Lignin has the highest decrement in the volume of crossover hydrogen. It must be remembered that this lignin is the most viscous solution used at the experiments (see Figure 3). Therefore, there may be a relation between the viscosity of the solution and the gas volume that crosses through the membrane, in which the higher the viscosity, the less the effect of the crossover phenomena. In addition, comparing the H_2_ produced at the cathode (shown in Figure 6a) with the data given in Figure 5, it has been demonstrated that a higher catalytic loading provides a better value for the higher potential, as expected. However, it does not imply an improvement on the H_2_ volume produced with both catalysts. In fact, it produces a worsening effect with a lower H_2_ production due to unknown reasons that have not been studied yet.

Figure 6b shows the total gas produced at the anode and its composition, which is analysed by mass spectrometry. It can be observed how the formed products are O_2_, CO, and H_2_. Stoichiometry is not fulfilled on the anode, as it does not produce the expected amount (~3.74 mmol h^−1^). Additionally, CO is detected due to degradation problems on the graphitic bipolar plates, which will be explained later.

For an easier comparison of the efficiencies on the total hydrogen generated, the main results are collected in Table 1.

Apart from the CP study, a CA study was carried out at −1.8 and −2.3 V (see Figure 7). As described by Caravaca et al. [37], and reported in many other works [34,35,36,38] at different conditions, lignin electrolysis always gives higher current densities for the same E applied for water electrolysis. In Figure 7, this is demonstrated for all the tests conducted: Lignin materials produce around 150% higher current densities at −1.8 V, applied at cathode, and around 35% higher at −2.3 V.

It is a fact that Pruning B. Lignin did not give the same current response as the commercial lignin. In contrast with the results obtained at the three-electrode cell, where Pruning B. Lignin was the material with the best response obtained, this does not occur at the flow cell. In this system, commercial lignin always gives the higher current density. It must be hypothesised that both systems differ from one another due to the operational mode. While the three-electrode cell system will depend on the mass transport, specifically in diffusion transport, the flow cell should minimise those effects. The flow received at the second system can lead to the disregard of the diffusion contribution, which is due to the forced convection conditions imposed by the effect of the solution movement through the electrode. Despite working at settled and fixed conditions, viscosity may be the reason for these results. The introduced flow may not be optimal due to the higher viscosities. This can lead the same active species to not be introduced into the system in both lignin tests. As previously demonstrated, the Pruning B. Lignin solution is around 53% more viscous over water viscosity, whereas the commercial lignin solution is 34% more viscous over water. This can be the reason for the less current response given by the Pruning B. Lignin. To prove this, we changed the flow rate when applying CA to check for a possible variation on the current response, as shown in Appendix A.

This test can ensure the proposed hypothesis of the effect of viscosity on the transported matter at a determined flux. Using a higher flux, from 23 to 27 mL min^−1^, an enhancement on the current response obtained can be observed. Moreover, a flux of 32 mL min^−1^ was tested and the same value for the limiting current density for 27 and 32 mL min^−1^ was obtained.

Moreover, the obtained gases at the cathode and anode were stored and measured. The result is shown in Figure 8a,b.

As it was expected, the electrolysis of lignin solutions provides higher density currents and higher cathodic production rates. At the anode, lignin generally diminishes the gas volume produced, indicating that in the best scenario, OER is being substituted by LOR. Comparing the results between −1.8 and −2.3 V, it can be seen that, at −1.8 V, the volumetric flow rate produced at the cathode is duplicated with the use of commercial lignin for all the catalytic loadings. At −2.3 V, an increase in gas production at the cathode is clear, but it is not quite evident in comparison with the results at −1.8 V. Therefore, the anode gas production seems to be significantly lower at −2.3 V than at −1.8 V. It must be considered that, for −1.8 V, the cathode gas product is duplicated with commercial lignin. The data are presented in Figure 8c,d.

The information obtained in the mass spectrometry study of the gas products of the CP can be extrapolated, qualitatively, to those obtained in the CA. It is expected by stoichiometry that the anode gas production must be 50% of the cathode gas production when NaOH solution is used. Supported by Figure 8a,b, it can be demonstrated that, in most cases, it does not occur, as in the event when CP tests were carried out (see Figure 6b). Looking at the ratio anode/cathode values of Figure 8c, only the water electrolysis (NaOH solution) at −1.8 V seems to fulfil the stoichiometry in most cases. As it occuredor the CP study, lignin solutions substitute the OER by the LOR, reducing the O_2_ production despite having a higher production of H_2_.

Figure 8d shows the winnings between the gas volume obtained at the cathode using lignin solutions with respect to a NaOH solution. Commercial lignin tends to be the material that produces the highest H_2_ volume when it is used as anolyte. Although Pruning B. Lignin does not produce the same H_2_ volume that commercial lignin does, it must be considered as the test realised in regard to flux and viscosity, in which it was demonstrated that the same flux for both solutions does not provide the same flow due to the viscosity difference between them. Moreover, it is observed that the higher increase in H_2_ flow rates is obtained at E = −1.8 V in comparison with water electrolysis. When higher polarisations are applied between the plates (−2.3 V), the H_2_ winnings are lower in comparison with the results at −1.8 V. In conjunction with the information presented in Figure 7, this denotes that LOR is more favoured than the less polarised OER.

As observed in Figure 8c, at −2.3 V, the stoichiometry is not fulfilled in the generated gas products for NaOH solution, as it was already seen at the CP (see Figure 6b). Moreover, the appearance of CO might indicate a corrosion or degradation problem on the anode graphitic bipolar plate that we confirmed after visual examination. The plate and the separator cloth were degraded as shown in Appendix A. For comparison, the undegraded cathode cloth and bipolar plate are shown in Appendix A.

The corrosion problem can be reasoned as follows: While the responses in the CP tests are more negative than E = −2.3 V, the CAs are performed at −2.3 V and do not fulfil the stoichiometry of the gas products. However, those CAs performed at E = −1.8 V seem to produce the expected volume at the anode. Therefore, it may be gathered that, from a more negative response than E = −1.8 V, the corrosion is a competitive reaction that seems to be favoured avoiding the OER. In fact, as stated by Yi et al. [59] and by *Busetti* et al. [60], the graphite can be highly corroded when immersed in a NaOH 1 M solution. In this line, some authors have described the use of carbon sacrificial anodes to substitute the OER [61,62,63]. This phenomenon occurs due to the easier path that is followed by the carbon oxidation in comparison with OER.

In agreement with the observations of Yi et al. [59], the electrode solution turns into a brownish colour after the realisation of CP for around 15 min. To check whether the NaOH solution was the main cause of this phenomenon, it was tested with a KOH 1 M solution. Several CPs were carried out with this new electrolyte. The gas products were measured and analysed by mass spectrometry. In Figure 9a, the responses of the different tests are shown. Figure 9b shows that the gas volume flow rate produced at the anode of the KOH tests, although it is not the stoichiometric one (102 mL h^−1^), is significantly more than the rate produced with NaOH as electrolyte. This may suggest that NaOH is significantly more corrosive and favours the graphite degradation. Furthermore, the gases obtained from the CP test of the KOH were analysed by mass spectrometry. Appendix A shows a comparison in the percentages of the gas composition with the different electrolytes.

The solutions from the anode were analysed by UV-Vis spectrometry. In Figure 10a, the spectra for both electrolytes are shown. In agreement with Yi et al. [59], an intense peak around 220 nm and a shoulder around 235 nm are found due to the presence of polycyclic aromatic hydrocarbons (PAHs). This is the confirmation for the suspected event. The graphite is suffering from the corrosion emitting this kind of compound to the solution.

It is demonstrated in Appendix A that the corrosion problem is enhanced when NaOH is used as anolyte due to the higher amount of CO detected at the anode gas product. It is almost four times higher in NaOH solution (4.3% CO with NaOH in contrast with the 1.2% obtained with KOH). Despite this, graphite corrosion is not solved with the use of KOH, since this problem is even present with this electrolyte. This can be proved with the UV-Vis spectra of both solutions after using it as anolyte. It can be seen in Figure 10a how in both spectra the afore-mentioned bands, described by Yi et al. [59], are present for the polycyclic aromatic hydrocarbons. In addition, as with both electrolytes, the stoichiometry is not fulfilled in the anode, and no CO_2_ is detected in higher percentages than 0.1%; therefore, it is thought that the graphite corrosion gives rise to dissolved CO_2_. This theory makes sense due to the high pH used at the solution (pH = 14). Then, the carbon that does not reach complete oxidation is detected in the form of CO gas and aromatic hydrocarbons in solution.

In conclusion, NaOH could be a good electrolyte due to its affordability and since it allows for lignin solubilisation. Despite the graphite degradation, it must be remembered that, as indicated in the CAs, the application of E = −1.8 V leads to a fulfilment of the stoichiometry of the generated gas. This would indicate that graphite corrosion of the bipolar plates should not be a main inconvenient. On the contrary, an applied voltage more negative than E = −1.8 V gives rise to a lessening on the anode production due to graphite corrosion. Moreover, the CO detection at the CP anode gas products may relate this to the diminishing anode production, as formerly explained.

In summary, as CO production is lowered when KOH is used, it also occurs when the lignin solution is added to the anode. Therefore, it may be gathered that CO is not produced due to the lignin material oxidation, but for the corrosion degradation of the graphite bipolar plate, which preferably occurs with NaOH 1 M solution at the anode. In fact, the lignin solution used as anolyte diminishes the CO production, which indicates that the LOR is preferred over OER and the graphite degradation.

### 3.5. Spectroscopic Characterisation of Lignins

#### 3.5.1. IR Characterisation

Lignin samples were observed to present different functional groups that can be distinguished in the normalised IR spectra of both dried materials presented in Appendix A. One of the main bands is located around 1700 cm^−1^ and assigned to the presence of carbonyl groups (-C=O). In Pruning B. Lignin spectrum, this functional group is identified by the appearance of a clear band around 1733 cm^−1^. However, for commercial lignin spectrum, this information it is not quite clear. It is not observed as an explicit band, but a shoulder around 1730 cm^−1^, merged with other bands indicating the presence of carbonyls in the material. Moreover, it is important to note that both lignins have in common the presence of a stretching band identified for two carbons connected by a double bond (-C=C-). For the commercial lignin, this band appears around 1585 cm^−1^, whereas for Pruning B. Lignin, this appears around 1609 cm^−1^. IR spectroscopy was also a determining factor to characterise the before-oxidation and after-oxidation steps and to state a conclusion about the capability of oxidising lignin. Figure 10b shows both IR spectra for commercial and Pruning B. Lignin (6 g L^−1^ in NaOH 1 M) before and after using these solutions. It can be appreciated how in the shown region the bands for the oxidised materials increase the relative absorbance, a clear sign of the increase in the stretching bands for -C-O bond and -C=C- bond after oxidation.

#### 3.5.2. NMR Characterisation

NMR spectra served as a support for completing the characterisation and to achieve deeper insights into the lignin functionalisation. Figure 11 summarises the NMR spectra for oxidised and non-oxidised commercial (see Figure 11a) and biomass (see Figure 11b) lignins. Both materials show a clear oxidation after H_2_ production experiments, as can be seen in the increase in the bands for oxidised groups as -O-CH_3_ and -CH=CH-CH_3_, also confirmed in the IR spectra. Moreover, in both lignins, the emergence of bands related to -C≡C-H group is observed, which denotes a high oxidation stage of the material. Commercial sample presents initially two different types of -CHO groups before oxidation. However, after the oxidation process, the selective formation of one of the -CHO groups is observed. Regarding Pruning B. Lignin, only one -CHO group was observed, and its NMR signal increases after the oxidation process. In addition, Pruning B. Lignin sample suffers a decrease in the aromatic groups after its use. All the presented clues drive to the demonstration that both materials clearly suffer in an oxidation process that provides the required electrons to the HER.

## 4. Conclusions

Electro-oxidation of lignin-based materials, employed as an additive in the anode in alkaline water electrolysis polymer electrolyte membrane reactor, demonstrated an effective way to avoid OER and to enhance the efficiency of H_2_ production. Less absolute potential is given as a response for lignin electro-oxidation when the same current is imposed for water electrolysis.

Furthermore, at E = −1.8 V, the lignin oxidation reaction (LOR) is more competitive with the OER than when E = −2.3 V is applied.

Higher current densities and higher H_2_ flow rates, with respect to water electrolysis, are achieved when E = −1.8 V is applied. When using commercial lignin as anolyte, Pt-Ru at 0.09 mg cm^−2^ as catalyst and applying −1.8 V, an increase of 200% in H_2_ flow rates with respect to water electrolysis was obtained. Moreover, the oxidation process was proved by structural characterisation, using infrared spectroscopy and ^1^H-NMR.

In addition, the viscous nature of the lignin solution is proved as the explanation for the H_2_ crossover through the membrane. Therefore, it is an additive that enhances the H_2_ production and allows for a higher yield of the HER and better control of the H_2_ crossover.

As a complementary test, NaOH 1 M was proved to produce a corrosion degradation on the graphite bipolar plates of the flow reactor when it is used as an additive for the electrolysis tests. Although KOH 1 M also produces a corrosion issue on the graphite, it is lower in comparison with the one produced by NaOH 1 M.

## Figures and Tables

**Figure 1 materials-16-03525-f001:**
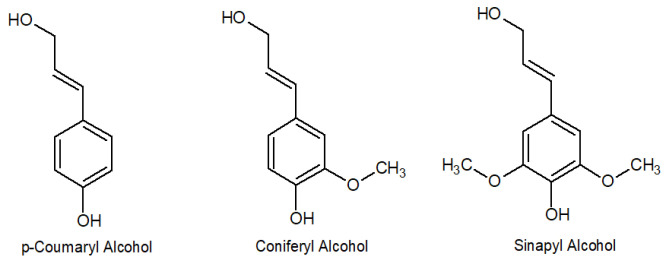
Three principal phenylpropanoids that constitute lignin.

**Figure 2 materials-16-03525-f002:**
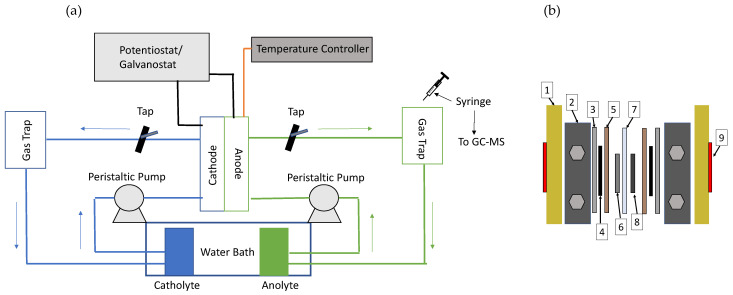
(**a**) Scheme of the complete set up. (**b**) Scheme of the polymer electrolyte membrane reactor flow cell. (1) Housing, (2) bipolar plate, (3) gasket, (4) carbon cloth separator, (5) gasket, (6) Ni foam cathode, (7) membrane, (8) spray-coated anode, (9) heating resistance.

**Figure 3 materials-16-03525-f003:**
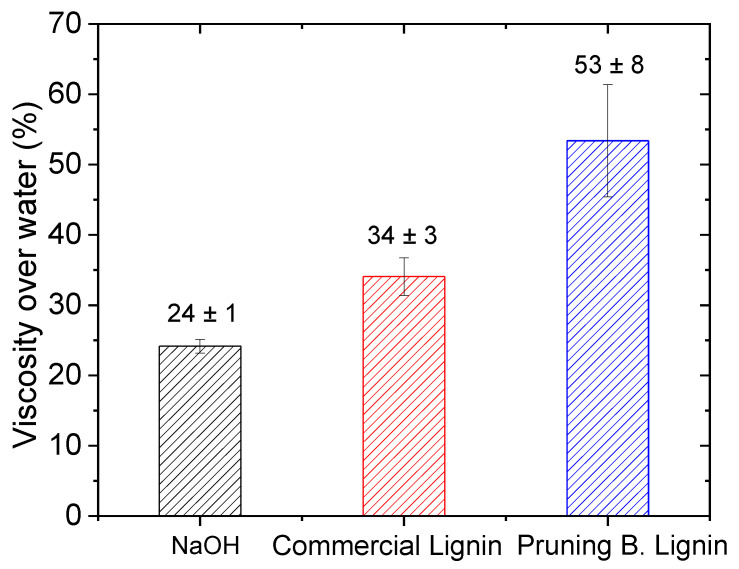
Viscosity comparison of the three different solutions, in percentage, with respect to water measured at 20 °C. Five measurements were obtained for each solution; the standard deviation is shown.

**Figure 4 materials-16-03525-f004:**
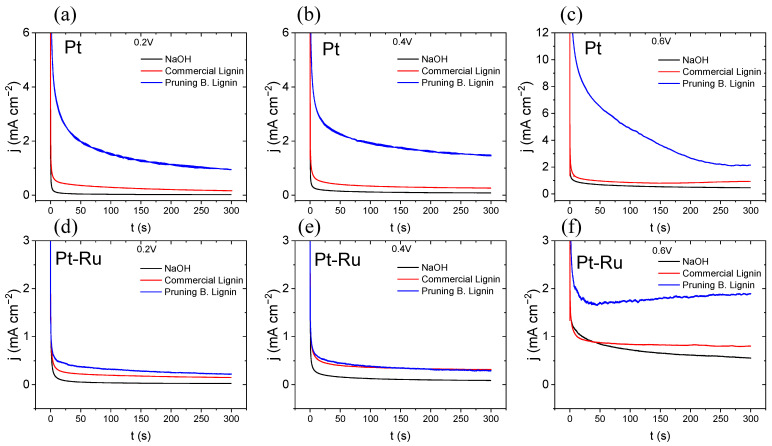
Chronoamperometries of lignin materials over both catalysts and at different potentials, 300 s, 60 °C, (**a**) with Pt at 0.2 V, (**b**) Pt at 0.4 V, (**c**) Pt at 0.6 V, (**d**) Pt at 0.2 V, (**e**) Pt at 0.4 V, (**f**) Pt-Ru at 0.6 V.

**Figure 5 materials-16-03525-f005:**
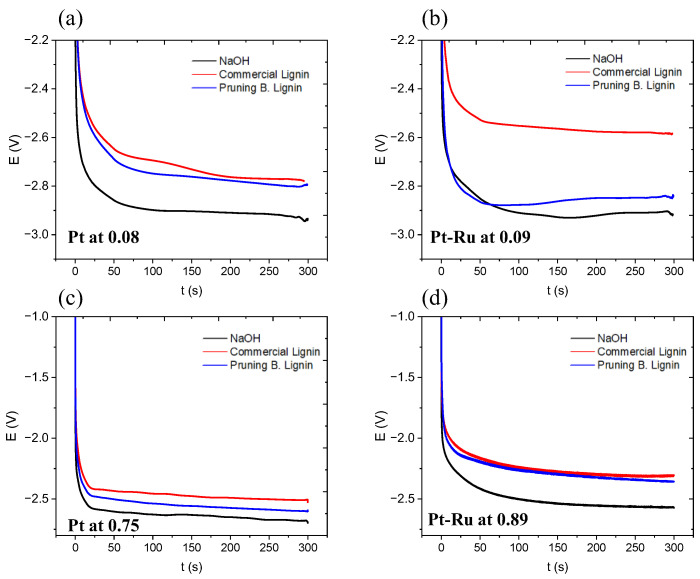
Chronopotentiometries at −100 mA cm^−2^, 300 s, 60 °C, (**a**) with Pt at 0.08 mg cm^−2^, (**b**) Pt-Ru at 0.09 mg cm^−2^, (**c**) Pt at 0.75 mg cm^−2^, (**d**) Pt-Ru at 0.89 mg cm^−2^.

**Figure 6 materials-16-03525-f006:**
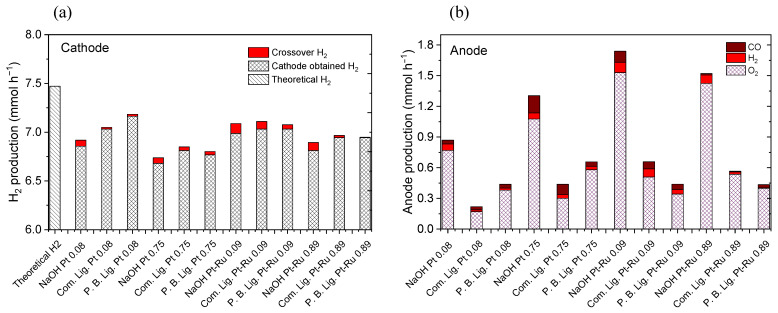
(**a**) Total H_2_ production in flow units (mmol h^−1^), (**b**) analysed gases produced at the anode in flow units (mmol h^−1^) from CP. Note that P.B. Lig. is Pruning B. Lignin. Two measurements for each quantification. Standard deviations are not shown in the graph for clarity, average obtained values are below 10% of the error in the quantification of CO_2_, CO, H_2_, and O_2_.

**Figure 7 materials-16-03525-f007:**
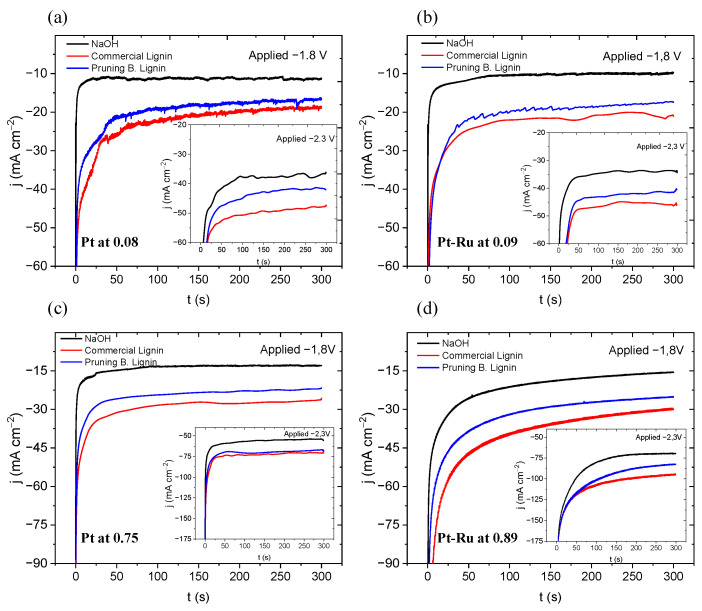
Chronoamperometries at −1.8 and −2.3 V, 300 s, 60 °C, (**a**) with Pt at 0.08 mg cm^−2^, (**b**) Pt-Ru at 0.09 mg cm^−2^, (**c**) Pt at 0.75 mg cm^−2^, (**d**) Pt-Ru at 0.89 mg cm^−2^.

**Figure 8 materials-16-03525-f008:**
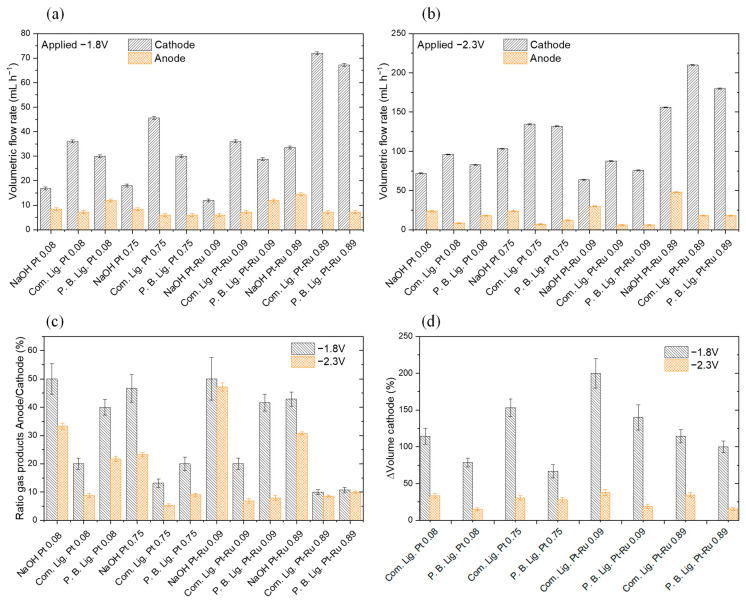
(**a**,**b**): Gas volumetric flow rates obtained at cathode and anode with different catalysts at different catalytic loadings. (**a**) Applying −1.8 V, (**b**) applying −2.3 V. (**c**,**d**): (**c**) Ratios of the gas products obtained as anode/cathode in percentage at −1.8 and −2.3 V; (**d**) volume variation, in percentage, produced at the cathode with respect to NaOH at −1.8 and −2.3 V. Note that P.B. Lig. is Pruning B. Lignin. Two measurements for each solution; standard deviation is shown.

**Figure 9 materials-16-03525-f009:**
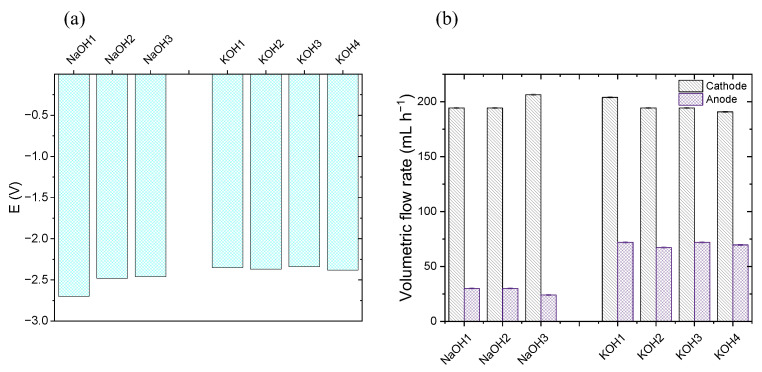
(**a**) Responses of the different tests at −100 mA cm^−2^, Pt at 0.08 mg cm^−2^, and 300 s; (**b**) volume of the gas products obtained at the cathode and anode. Two measurements for each solution; the standard deviation is shown.

**Figure 10 materials-16-03525-f010:**
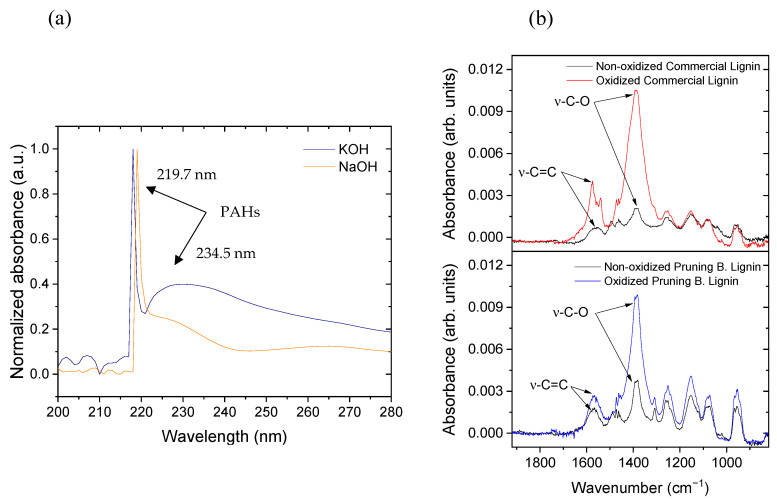
(**a**) UV-Vis spectra of the used anolyte; (**b**) IR spectra of oxidised and non-oxidised lignin in solution.

**Figure 11 materials-16-03525-f011:**
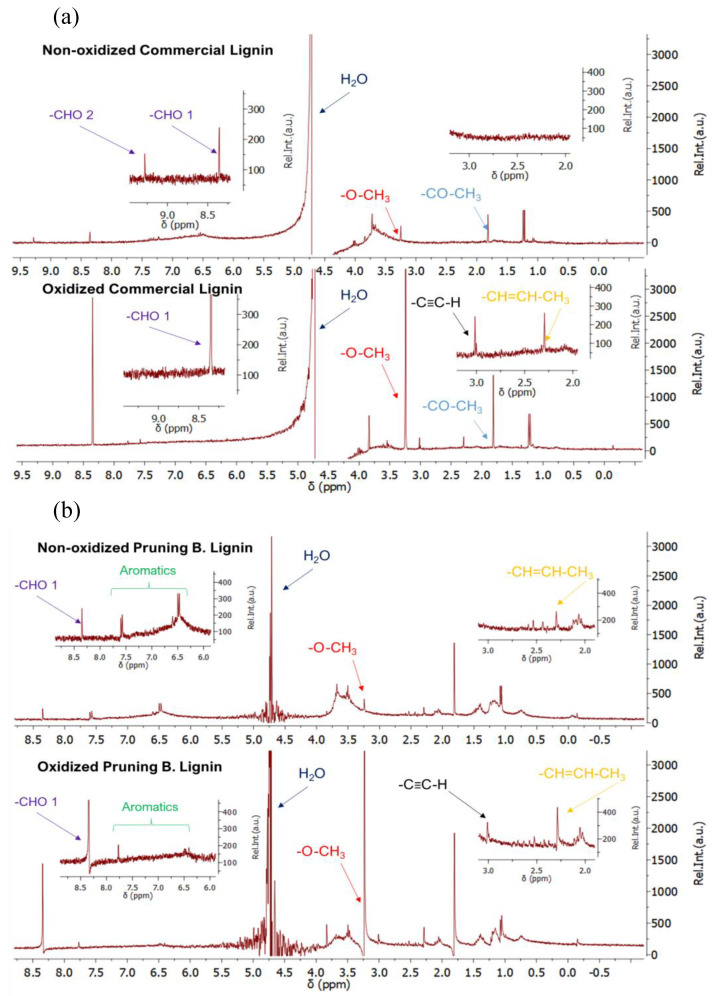
NMR characterisation for non-oxidised (top) and oxidised (bottom); (**a**) commercial lignin, (**b**) Pruning B. Lignin.

**Table 1 materials-16-03525-t001:** Efficiencies in percentage of total H_2_ produced (obtained at the cathode + crossover losses) and H_2_ obtained directly at the cathode from CP.

		Total H_2_ Efficiency (%)	H_2_ Efficiency on the Cathode (%)
Pt 0.08	NaOH	92.8	91.8
	Commercial Lig.	95.1	94.1
	Pruning B. Lig.	96.9	95.9
Pt 0.75	NaOH	90.4	89.4
	Commercial Lig.	92.2	91.2
	Pruning B. Lig.	91.6	90.6
Pt-Ru 0.09	NaOH	94.5	93.5
	Commercial Lig.	95.1	94.1
	Pruning B. Lig.	95.1	94.1
Pt-Ru 0.89	NaOH	92.2	91.2
	Commercial Lig.	93.9	92.9
	Pruning B. Lig.	93.9	92.9

## Data Availability

Not applicable.

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
