# Peer review of "Clean H2 Production by Lignin-Assisted Electrolysis in a Polymer Electrolyte Membrane Flow Reactor"

_materials, 2023, doi:10.3390/ma16093525_

Round 1
Reviewer 1 Report
This manuscript reported the efficient H2 production by the substitution of the OER with lignin materials in water splitting. Although the manuscript looks rich, the article lacks innovation/scientific significance. In addition, some of the language/description are not standard, which makes it difficult for readers. Considering the current status of the submitted manuscript, it is suggested to reject.

Extensive editing of English language need to be required.
Author Response
Answers to Reviewer 1
Comments and Suggestions for Authors
This manuscript reported the efficient H2 production by the substitution of the OER with lignin materials in water splitting. Although the manuscript looks rich, the article lacks innovation/scientific significance. In addition, some of the language/description are not standard, which makes it difficult for readers. Considering the current status of the submitted manuscript, it is suggested to reject.
A: Thank you very much for the review and your comments. We will improve the manuscript taking into account all your comments and we hope that the new version will be acceptable to you for publication in Materials. Details about the changes introduced in the manuscript and specific responses to each of the comments are provided in the Annex below. The amended text, added and removed, is highlighted in red colour in the revised manuscript for direct inspection.
Comments on the Quality of English Language
Extensive editing of English language need to be required.
A: Thank you very much for the review and your comments. We will thoroughly check the English of the manuscript with the help of fluent English speaker colleagues or with the help of mpdi corrections. We would like to convey to you that the other two reviewers have found an acceptable standard written level and they only propose minor English changes. Please, see revised version showing changes. We hope that the new version will be acceptable to you for publication in Materials.
- In the section of lntroduction, the description of the background is too cumbersome. On the contrary, the differences between this work and the previous work need to be further emphasized.
A: As mentioned before, we have carefully revised the English. In addition, we have reduced the introduction and emphasized more the current state of the art, as well as our contribution. Please, see revised version showing changes.
- The paper points out that Caravaca/Lalvani/Khalid et al. have studied this system (the use of lignin as an additive on the anode of an electrolyser in alkaline process). However, the authors choose two different lignin materials, one commercial and other from pruning biomass. Moreover, the corresponding electrode materials are also What is the innovation or scientific significance of the article?
A: Thank you very much for the review and your comments. In our opinion, the scientific contribution of our work is focused on the development of an electrolyser (at laboratory scale) that uses biomass from the community of Madrid, in line with the current strategic actions in Europe focused on circular economy models. In addition, we propose the use of low-cost anion exchange membranes. The proposed catalysts are reasonably low cost and, especially in the anode, where the use of Pt and Pt/Ru is done with quite low catalytic loadings (much lower than in previous articles). Furthermore, Ni foam is used for cathodic reaction. This catalysts assembly, anode/cathode has not been reported yet in bibliography, as well as chronopotentiometric tests for lignin electro-oxidation in a flow cell, that has not been published yet. In fact, revising bibliography, Caravaca/Lalvani/Khalid et al. have not performed fix current experiments, nor they have not presented a precise characterisation of the gas products generated at the anode. Taking into account your comments we have added an explicit paragraph in the introduction. Please, see revised version showing changes.
- In the results and discussion section, most of the expression stays in the picture. Sorne of the necessary discussions need to be emphasized. For example, what exactly is the relationship between viscosity and its properties?
A: We have emphasized more on the discussion section, as for example the relationship between viscosity and properties in the new version of the manuscript. Please, see revised version showing changes.
- There are too many figures in the main text, and sorne unimportant figures should be moved to SI.
A: Thanks for the comment. Indeed, our manuscript contains 17 figures, and it seems like a high number. We have merged some figures and sent others to the supplementary material, leaving a total now of 11 figures in the main text.
- In Figure 16, why does the -C=C- bond increase after oxidation?
A: Lignins are complex polymers and the irregular structure that depends on its origin and extraction method. For this reason, the partial oxidation of lignin has allowed us to identify the main functional groups generated after electrolysis. Based on the experimental data, it does not appear that lignin has undergone to complete oxidation to small organic molecules. In fact, the main oxidized functional groups that have been formed and that we have identified by IR and NMR involve C=C, C-O and C=O bonds. That is, oxidative dehydrogenation processes (formation of C=C groups, including triple bonds), formation of aldehydes, ketones or ethers (C=O and C-O) that have partially oxidized the polymer.
- sorne of the descriptions make readers difficult, such as:
- "The development of society implies significant economic growth, currently making the economy of developed and developing countries highly dependent on fossil fuels, and therefore, on combustion processes that generate polluting products that ultimately increase the C02 concentration, among other products, well above the values thathave been naturally generated during the last thousands of years"
A: Thanks for the comment. We have simplified this paragraph in a more understandable way. Please, see revised version showing changes.
- "In fact, as it could be appreciated at the Fig 12a; where the anode gas products do not reach the total efficiency (50% of the gas products of the cathode due to stoichiometry), this neither happens to the results obtained at most of the CA (Figs 11a and 11b)"
A: We have clarified this explanation and rewritten this paragraph. In fact, in the water splitting reaction, the oxygen production must be the half in comparison to the hydrogen production according to stoichiometry of the reaction. Experimentally, this is not seen at most of the experiments performed, in which it was discovered that carbon oxidation from the bipolar plates occurs while electrolysis tests are being run.
Added paragraph:
“It is expected by stoichiometry that the anode gas production must be 50% of the cathode gas production, when NaOH solution is used. Supported by Figures 8a and 8b, it can be demonstrated that in most cases it does not occur, as well as it happened when CP tests were being run (see Figure 6b).”
And
“Attending to Figure 8c, at -2.3V it is not seen a fulfilment of the stoichiometry in the generated gas products for NaOH solution, as it was already seen at the CP (see Figure 6b). Moreover, the appearance of CO made the suspect of a corrosion or degradation problem on the anode graphitic bipolar plate. It was a fact when this piece was examined in an accuracy way. The plate and the separator cloth were on the path to be destroyed.”
- "In this IR characterization is important to remark the presence of carbonyls (-C=O) as a clear band around 1733 cm-1, for biomass lignin, andas a hidden shoulder around 1730 cm-1 for commercial lignin."
A: Lignin materials used in this work has been characterised by IR spectroscopy among others characterization techniques as NMR. It has been found that these materials present carbonyl groups in its own structure. This is important to know because this is an oxidised group that is observed to be increased after the electro-oxidation. This paragraph has been rewritten and readapted in a simpler way to make it more understandable.
Added paragraph:
“Lignin samples were observed to present different functional groups among them. The normalized spectra of both dried materials are presented in Figure S5. One of the main bands is located around 1700cm-1 and assigned to the presence of carbonyl group (-C=O). In Pruning B. Lignin spectrum, this functional group is identified by the appearance of a clear band around 1733cm-1. However, for commercial lignin spectrum, this information it is not so clear. It is not observed an explicit band, but a shoulder appears around 1730cm-1, merged to another band indicating the presence of carbonyls in the material.”
- Sorne detail errors should be further corrected, such as the space between the number and the unit, Journal Name in References.
A: We have checked all, thank you for the remark.

Reviewer 2 Report
The research topic on practical water splitting is very interesting. Moreover, abundant experiments have been performed by the authors. Nevertheless, there are still some major issues need to be further addressed in this contribution. Thus, I recommend the manuscript to be accepted after the following revisions:
1. What is the accurate definition of the voltage, potential and/or E in the manuscript?
2. The authors should provide some basic morphology and (micro)structural characterizations for the catalysts before and after catalysis, at least for the best sample.
3. Why was the carbon cloth selected as the anode substrate? It is commonly accepted that the carbon cloth can be decomposed at large OER potential.
4. More references on water splitting reported in the recent years should be included in this manuscript to show the importance of this field.
5. The language and details should be greatly checked and improved.
The language should be improved to a large extent.
Author Response
Answers to Reviewer 2
The research topic on practical water splitting is very interesting. Moreover, abundant experiments have been performed by the authors. Nevertheless, there are still some major issues need to be further addressed in this contribution. Thus, I recommend the manuscript to be accepted after the following revisions:
A: Thank you very much for the review and your comments. We will improve the manuscript taking into account all your comments and we hope that the new version will be acceptable to you for publication in Materials. Details about the changes introduced in the manuscript and specific responses to each of the comments are provided in the Annex below. The amended text, added and removed, is highlighted in red colour in the revised manuscript for direct inspection.
- What is the accurate definition of the voltage, potential and/or E in the manuscript?
A: Thank you very much for the review and your comments. We have checked and unified the voltage, potential and/or E definition in the manuscript. Please, see revised version showing changes.
- The authors should provide some basic morphology and (micro)structural characterizations for the catalysts before and after catalysis, at least for the best sample.
A: Thank you very much for the review and your comments. The reviewer makes an important comment which is the characterization of the catalyst before and after the LOR. It is our work, the operation time has been short (300 s) since we worked at laboratory-scale experiments to characterize the Pruning lignins of the community of Madrid. It is necessary to point out that the catalysts are commercial and have not been synthesised in our laboratory. The next step, once the lignins to be used are selected, would be the characterization of the electrolyser at long operating times (hours, even days), especially after use in order to check the final state and analyse the possibility of reusing it. All the parts of the flow cell is very relevant and one of those elements is undoubtedly the catalyst.
- Why was the carbon cloth selected as the anode substrate? It is commonly accepted that the carbon cloth can be decomposed at large OER potential.
A: Thank you very much for the review and your comments. The reviewer makes an important comment which is the observed degradation issue in the carbon cloth. The anodic nanoparticulate materials used in this work are based on Pt or Pt-Ru that are already supported on carbon vulcan (commercial compounds). Once the ink was prepared following the procedure described in our work, they were sprayed on a hydrophilic carbon cloth to have a homogeneous distribution of the catalyst in the anode. That is, the structure of interlocked fibers of the carbon cloth helps in the process of homogenization of the sprayed ink that contains the catalyst mass that we want to use. In this way, we facilitate the access of the active species (commercial and non-commercial lignins in our case), which are much larger than OH- ions and generate more viscous solutions than NaOH or KOH. As commented in the manuscript, the electrolysis at the highest fixed potential studied by us (-2.3V), leads the degradation of the carbon cloth and the graphite bipolar plate as we explained in the discussion of results. We would like to address this issue in future works by changing the catalyst supports or even changing the graphite bipolar plates to deeper study the effects in the electrolysis process.
- More references on water splitting reported in the recent years should be included in this manuscript to show the importance of this field.
A: Thank you for this interesting comment. Following the reviewer's advice, we have improved the introduction by referencing the most recent works in the lignin-assisted electrolysis literature and AEM technology. We have added the following paragraph and references:
“In recent years, several advances have been made in AEM electrolysis. Recently, new catalysts [1–3] such as stainless steel and Ni-Fe alloys are being studied for this purpose. The development of new membranes [4,5] and different technoeconomic studies [6,7] generally show a clear desire of the scientific community to advance in the AEM process for the production of H2.
- Martinez-Lazaro, A.; Caprì, A.; Gatto, I.; Ledesma-García, J.; Rey-Raap, N.; Arenillas, A.; Espinosa-Lagunes, F.I.; Baglio, V.; Arriaga, L.G. NiFe2O4 Hierarchical Nanoparticles as Electrocatalyst for Anion Exchange Membrane Water Electrolysis. Journal of Power Sources 2023, 556, 232417, doi:10.1016/j.jpowsour.2022.232417.
- Ďurovič, M.; Hnát, J.; Strečková, M.; Bouzek, K. Efficient Cathode for the Hydrogen Evolution Reaction in Alkaline Membrane Water Electrolysis Based on NiCoP Embedded in Carbon Fibres. Journal of Power Sources 2023, 556, 232506, doi:10.1016/j.jpowsour.2022.232506.
- Nuggehalli Sampathkumar, S.; Ferriday, T.B.; Middleton, P.H.; Van Herle, J. Activation of Stainless Steel 316L Anode for Anion Exchange Membrane Water Electrolysis. Electrochemistry Communications 2023, 146, 107418, doi:10.1016/j.elecom.2022.107418.
- Akbar Trisno, M.L.; Dayan, A.; Ji Lee, S.; Egert, F.; Gerle, M.; Rykær Kraglund, M.; Oluf Jensen, J.; Aili, D.; Roznowska, A.; Michalak, A.; et al. Reinforced Gel-State Polybenzimidazole Hydrogen Separators for Alkaline Water Electrolysis. Energy & Environmental Science 2022, 15, 4362–4375, doi:10.1039/D2EE01922A.
- Lv, B.; Yang, Y.; Yang, C.; Huang, Z.; Zhou, Y.; Song, W.; Hao, J.; Shao, Z. Layered Double Hydroxide Composite Membrane for Advanced Alkaline Water Electrolysis. International Journal of Energy Research 2022, 46, 11892–11902, doi:10.1002/er.7955.
- Naqvi, S.A.H.; Taner, T.; Ozkaymak, M.; Ali, H.M. Hydrogen Production through Alkaline Electrolyzers: A Techno-Economic and Enviro-Economic Analysis. Chemical Engineering & Technology 2023, 46, 474–481, doi:10.1002/ceat.202200234.
- Cost-Competitive Green Hydrogen: How to Lower the Cost of Electrolysers? Oxford Institute for Energy Studies.”
- The language and details should be greatly checked and improved.
A: Thank you very much for the review and your comments. We will thoroughly check the English of the manuscript with the help of fluent English speaker colleagues or with the help of mpdi corrections. We would like to convey to you that the other two reviewers have found an acceptable standard written level and they only propose minor English changes. Please, see revised version showing changes. We hope that the new version will be acceptable to you for publication in Materials.

Reviewer 3 Report
This article presents a detailed analysis of the lignin electro-oxidation process in a flow reactor and provides optimization strategies to improve the efficiency of the process. This research is a significant step towards the industrial implementation of the lignin electro-oxidation process, given the pre-industrial approach of the system. The findings of this study highlight the potential of lignin as an affordable and efficient additive for H2 production, which could have significant implications for the renewable energy industry. Some issues must be solved before it is considered for publication. If the following problems are well-addressed, this reviewer believes that the essential contribution of this article is vital for water electrolysis.
1. Is it possible to use other catalysts be used instead of Pt-Ru/C or Ni foam/particles for lignin electro-oxidation? How do they compare in terms of efficiency and cost?
2. How does lignin electro-oxidation compare to other methods of H2 production in terms of cost, efficiency, and environmental impact?
3. How can the lignin electro-oxidation process be scaled up for industrial implementation, and what are the challenges and limitations?
N/A
Author Response
Answers to Reviewer 3
Comments and Suggestions for Authors
This article presents a detailed analysis of the lignin electro-oxidation process in a flow reactor and provides optimization strategies to improve the efficiency of the process. This research is a significant step towards the industrial implementation of the lignin electro-oxidation process, given the pre-industrial approach of the system. The findings of this study highlight the potential of lignin as an affordable and efficient additive for H2 production, which could have significant implications for the renewable energy industry. Some issues must be solved before it is considered for publication. If the following problems are well-addressed, this reviewer believes that the essential contribution of this article is vital for water electrolysis.
A: We appreciate the positive comments and suggestions of the reviewers. We believe that the revised manuscript accounts for the different issues raised in the evaluation reports. Details about the changes introduced in the manuscript and specific responses to each of the comments are provided in the Annex below. The amended text, added and removed, is highlighted in red colour in the revised manuscript for direct inspection.
- Is it possible to use other catalysts be used instead of Pt-Ru/C or Ni foam/particles for lignin electro-oxidation? How do they compare in terms of efficiency and cost?
A: Other catalysts have been reported to be effective for lignin electro-oxidation. In fact, IrO2-base oxide electrodes have been reported by Tolba et al. (10.1016/j.jelechem.2009.12.013) to be effective for lignin electro-oxidation. Moreover, Ti/Sb-SnO2 and Ti/PbO2 are other electrodes that are reported by Shao et al. (10.1016/j.cej.2014.01.074) to be effective for this purpose Although these electrodes, iridium oxide-based and titanium-based catalysts, have been demonstrated as effective, these examples proposed hereby have not been tested in a flow cell or electrolyser. This is a different experimental set-up that could provide results not directly comparable with ours because their system is based on a three-electrode cell. Of course, the development of efficient catalysts is a challenge for this type of electrolysers and it is necessary to go deeper into this point.
As comment before, direct comparison between these examples and our work may be risky, especially due to the different experimental set-up. For example, in our work we have found significant differences between the operation of an electrode in the absence of flow (three electrode cell) and in its presence (flow cell). Therefore, in a multivariable problem, the flow rate seems a relevant parameter to optimize. From the cost point of view, we would like to highlight that two of the most relevant factors in the final cost of a flow cell-based electrolyser would be the membrane and the catalysts.
- How does lignin electro-oxidation compare to other methods of H2production in terms of cost, efficiency, and environmental impact?
A: Thanks for this interesting question. The main H2 production methods can be classified as: Hydrocarbon reforming, Water electrolysis, Coal Gasification, Thermolysis, Biochemical Production, Photocatalysis and Photoelectrolysis. Steam reforming of natural gas is currently the most widely used process to produce hydrogen, although it has a high environmental impact as it uses fossil resources and emits a lot of CO2. The electrolysis of water is a cleaner alternative for the generation of H2 since it uses water as a reagent and if the electrolyser is powered by renewable energy (solar, wind, etc.), green H2 can be obtained. The great advantage of this proposal is the non-use of fossil sources and the non-emission of CO2. However, electrolyser technology is currently not yet competitive if we compare it with gas reforming, especially for the price per Kg of H2 obtained. It is true that great improvements are being made to make the electrolysis process more efficient and reduce its cost. An example is the use of additives in the anode (lignins), the use of cheaper membranes (anionic, for example) or the search for low cost catalysts or those that require small catalytic charge. Moreover, it must be pointed out that in many other previous works where other organic materials try to substitute the OER reaction, as for example with alcohols-based additives which are more expensive than lignins. It must be reminded that lignin is a residue from paper industry with a scarce industrial application. On interesting alternative for the re-utilization of this residue is the above mentioned use as anode additive in electrolysers.
- How can the lignin electro-oxidation process be scaled up for industrial implementation, and what are the challenges and limitations?
A: The industrial scaling of this technology is not easy and it is not sufficiently mature at this stage over the PEM electrolyser and the high temperature solid state electrolyser. Commercial electrolysers already exist and the goal would be to improve their efficiency by modifying the anode solution. From a circular economy point of view, the lignin becomes from the pruning residues that, properly treated, would obtain the lignin fraction (actually without industrial application). From a more engineering point of view, the challenges faced by this technology range from the homogeneity of the membrane and catalysts, mechanical stability at long operating times, pumping large amounts of solution, to problems derived from degradation at alkaline pH. Furthermore, a very important issue is the crossover of H2 to the anode, since a high percentage would even
(x) Minor editing of English language required
A: Thank you very much for the review and your comments. We will thoroughly check the English of the manuscript with the help of fluent English speaker colleagues or with the help of mpdi corrections.

Reviewer 4 Report
The authors presented some interesting work on using lignin for hydrogen production. However, it seems that the authors did not identify the by-products from lignin oxidation. It would be interesting to know the product of the reaction. I recommend for publication after the following has been addressed.
L115-116, are the metal loadings in mol% or wt%?
L127, how were the catalyst losses estimated? what assumptions were made?
L129, details of the "catalyst spraying" were not provided. did the author used a manual air brush for spraying? how did the author dry the sample?
L158, it is not clear what the author's intention was. Was the author trying to test the reaction at neutral pH? was the author trying to test the effect of acidic medium?
inconsistent use of American and British English spellings.
Fig 5a,b and Fig 6a,b,c should be replotted either as a combined graph or with the same scale for better comparison.
How did the author determine the amount of hydrogen crossover?
The reactor setup is rather ambiguous. How did the author prevent gas build up in the anode and cathode chamber? From the figure, there seem to be some overhead space for trapping gas.
minor editing required. inconsistent use of American and British English spellings.
Author Response
Answers to Reviewer 4
Comments and Suggestions for Authors
The authors presented some interesting work on using lignin for hydrogen production. However, it seems that the authors did not identify the by-products from lignin oxidation. It would be interesting to know the product of the reaction. I recommend for publication after the following has been addressed.
A: We appreciate the positive comments and suggestions of the reviewers. We believe that the revised manuscript accounts for the different issues raised in the evaluation reports. Details about the changes introduced in the manuscript and specific responses to each of the comments are provided in the Annex below. The amended text, added and removed, is highlighted in red colour in the revised manuscript for direct inspection.
Concerning oxidation products of lignins, it is important to highlight that lignins are complex polymers and the irregular structure that depends on its origin and extraction method. For this reason, the partial oxidation of lignin has allowed us to identify the main functional groups generated after electrolysis. Based on the experimental data, it does not appear that lignin has undergone complete oxidation to small organic molecules. In addition, our main objective was to obtain H2 in greater quantity and at a lower cell potential. For this reason, we try to characterize as well as possible the oxidation process at the anode. Finally, a deeper characterization of the oxidized structure could be covered with mass spectrometry techniques that are widely used for the characterization of polymers. Even with the mass spectrum, a structural identification of the polymer would be a very complex task since lignin (oxidized or not) presents very different monomeric units. There are other possibilities such as the 2D MNR that provided more structural information.
L115-116, are the metal loadings in mol% or wt%?
A: Metal nanoparticles are in wt%, it has been checked and modified in the manuscript. Thanks for the comment.
L127, how were the catalyst losses estimated? what assumptions were made?
A: During different tests it was observed that in the spraying process it was lost part of the sprayed material. Due to the manual nature of the procedure, a 20% of catalytic losses is assumed during the cloth spraying, for that reason, catalyst nanoparticles are overestimated in weight when the ink is formed before being sprayed. Possible sources of loss may be: i) the ink that remains in the airbrush; ii) part of the sprayed ink is deposited in areas outside the marked area that we will use as a catalyst, etc. However, the cloth with the catalyst is weighed before and after the spray process, when the solvent has evaporated, and the sample has a constant weight thus giving the total catalyst weight per cm2.
L129, details of the "catalyst spraying" were not provided. did the author used a manual air brush for spraying? how did the author dry the sample?
A: The catalytic ink is sprayed with a manual airbrush; this cloth is placed over a temperature-controlled heating plate at 60°C to accelerate the drying process. After spraying, the carbon cloth is left over the heating plate around 30 minutes to ensure that it is dry. An explicit explanation has been added to the new version of manuscript.
L158, it is not clear what the author's intention was. Was the author trying to test the reaction at neutral pH? was the author trying to test the effect of acidic medium?
A: We wanted to test the solubility of lignins in a neutral and basic medium. In fact, our proposal is an electrolyser in a basic medium with an anionic membrane. The problem is that too alkaline a pH will cause degradation in the flow cell. In any case, we did the solubility test as part of the physico-chemical characterization. It is important to highlight that the non-commercial lignin is not characterized before, and a good solubility at less alkaline pH would allow proposing electrolysers working at lower pH that could suffer less degradation problems.
inconsistent use of American and British English spellings.
A: Thank you very much for the review and your comments. We will thoroughly check the English of the manuscript with the help of fluent English speaker colleagues or with the help of mpdi corrections.
Fig 5a,b and Fig 6a,b,c should be replotted either as a combined graph or with the same scale for better comparison.
A: Checking the figures and testing the commented suggestion, using the same scale for Fig 5a,b would make the voltammogram unreadable due to the different intensity in its peaks. The wanted purpose from this figure was to show an electrochemical characterisation of the two different lignins and to make a brief comparison between both lignins. As it is said, different scales for a and b will hide some important information. With the aim of reducing the number of figures, this 5 a, b was moved to the supporting information. Something similar occurs with figure 6. If it is observed at figure 6 c, the current response for Pruning B. Lignin is so high, especially on the transient region of the curve, that the replotting of this figure adapting the Y axis to the 6a and 6b scale will produce the complete invisibility of great part of the Pruning B. Lignin curve. Moreover, if 6a and 6b are tried to be rescale to 6c Y axis scale, no difference will be observed between lignin curves and NaOH curve. We believe that in this way it helps the reader to visualise the differences.
How did the author determine the amount of hydrogen crossover?
A: The H2 crossover percentage at the anode has been determined by GC-MS analyzing the gases generated at the anode. A gas syringe was used to take a sample from the gas traps (anode and cathode) and to introduce it into the chromatograph. Previously, a calibration curve was prepared with different gases; CO, CO2, and H2 , using Ar as diluter gas. Interpolating the intensities registered by the mass spectrometer into the calibration curved, it is determined and calculated the proportion of the different gases produced: i) H2 in the cathode; ii) O2, CO (partial oxidation of bipolar plate and carbon cloth) and H2 (Crossover from the cathode to the anode through the membrane.
The reactor setup is rather ambiguous. How did the author prevent gas build up in the anode and cathode chamber? From the figure, there seem to be some overhead space for trapping gas.
A: If we have correctly understood the question, the gases generated at the anode and cathode are taken to the gas traps thanks to the flow of the anolyte and catholyte solutions. These gases traps are previously filled with solution from the anode or cathode by means of a tap system and with the help of the flow. Finally, it should be noted that it is a sealed system that does not allow the entry of outside air and once the gas traps have been filled with solution, the gases are analysed by GC-MS.
The two taps after the gas traps are used to empty the air from the system and fill the traps with the generated gases. Firstly, when the taps are closed, the gas traps are purged letting the present air to escape out through a needle that is puncturing the sealing septum. While this is occurring, the flowing solutions start to fulfil the space that the air was occupying before. Once the solutions reach the top of the gas traps, the needles are removed, and the taps are opened to fill it with the generated gases. It is important to remind that it is a hermetic set up and no air can penetrate into the system nor gas products can escape out of the gas traps. In the new figure 2, the taps in the system have been included, which by mistake were not included in the old figure.
Comments on the Quality of English Language
minor editing required. inconsistent use of American and British English spellings.
A: Thank you very much for the review and your comments. We will thoroughly check the English of the manuscript with the help of fluent English speaker colleagues or with the help of mpdi corrections.

Round 2
Reviewer 1 Report
- The authors answer my questions well and I recommend acceptance.
-
The language problem has been corrected.
Author Response
A: We appreciate the positive comments and suggestions of the reviewers. We believe that the revised manuscript accounts for the different issues raised in the evaluation reports.

Reviewer 2 Report
I think the revised version of manuscript can be accepted now.
Author Response

(The authors gave the same response as above.)
